# Mitigating Data Imbalance and Representation Degeneration in Multilingual Machine Translation

**Wen Lai**[1,2]**, Alexandra Chronopoulou**[1,2]**, Alexander Fraser**[1,2]

[1]Center for Information and Language Processing, LMU Munich, Germany
[2]Munich Center for Machine Learning, Germany
{lavine, achron, fraser}@cis.lmu.de

## Abstract

Despite advances in multilingual neural machine translation (MNMT), we argue that there are still two major challenges in this area: *data imbalance* and *representation degeneration*. The data imbalance problem refers to the imbalance in the amount of parallel corpora for all language pairs, especially for long-tail languages (i.e., very low-resource languages). The representation degeneration problem refers to the problem of encoded tokens tending to appear only in a small subspace of the full space available to the MNMT model. To solve these two issues, we propose **Bi-ACL**, a framework which only requires target-side monolingual data and a bilingual dictionary to improve the performance of the MNMT model. We define two modules, named bidirectional autoencoder and bidirectional contrastive learning, which we combine with an online constrained beam search and a curriculum learning sampling strategy. Extensive experiments show that our proposed method is more effective than strong baselines both in long-tail languages and in high-resource languages. We also demonstrate that our approach is capable of transferring knowledge between domains and languages in zero-shot scenarios[1].

## 1 Introduction

Multilingual neural machine translation (MNMT) makes it possible to train a single model that supports translation from multiple source languages into multiple target languages. This has attracted a lot of attention in the field of machine translation (Johnson et al., 2017; Aharoni et al., 2019; Fan et al., 2021). MNMT is appealing for two reasons: first, it can transfer the knowledge learned by the model from high-resource to low-resource languages, especially in zero-shot scenarios (Gu et al., 2019; Zhang et al., 2020a); second, it uses

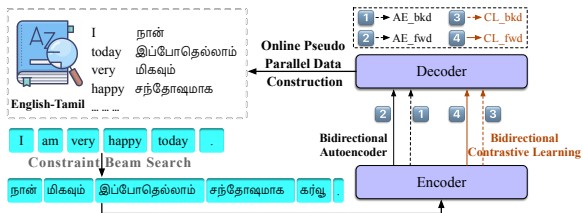

Figure 1: Method Overview. Our approach mainly consists of three parts: online constrained beam search, bidirectional autoencoder and bidirectional contrastive learning. Our approach explores the scenarios of using only target-side monolingual data and a bilingual dictionary to simultaneously alleviate the data imbalance and representation degeneration issues in large-scale MNMT model.

only one unified model to translate between multiple language pairs, which saves on training and deployment costs.

Although significant improvements have been made recently, we argue that there are still two major challenges to be addressed: i) MNMT models suffer from poor performance on long-tail languages (i.e., very low-resource languages), for which parallel corpora are insufficient or non-existing. We call this the *data imbalance* problem. For instance[2], 21% of the language pairs in the *m2m_100* model (Fan et al., 2021) have a BLEU score of less than 1 and more than 50% have a BLEU score of less than 5. Only 13% have a BLEU score over 20. For example, the average BLEU score for the language pairs with *Irish* as the target language is only 0.09. ii) Degeneration of MNMT models stems from the anisotropic distribution of token representations, i.e., their representations reside in a narrow subset of the entire space (Zhang et al., 2020b). This is called the *representation degeneration* problem. It can lead to a prevalent issue in large-scale MNMT: the model copies sentences from the source sentence or translates them into the wrong language (off-target problem; Zhang et al.,

---

[1]Our source code is available at https://github.com/lavine-lmu/Bi-ACL

[2]Please refer to Figure 3 of Appendix D.1 for more details.

2020a).

To address the *data imbalance* problem, prior work has attempted to improve the performance of a machine translation model without using any parallel data. On the one hand, unsupervised machine translation (Lample et al., 2018a,b) attempts to learn models relying only on monolingual data. On the other hand, bilingual dictionaries have shown to be helpful for machine translation models (Duan et al., 2020; Wang et al., 2022). What these approaches have in common is that they only require data that is both more accessible and cheaper than parallel data. As an example, 70% of the languages in the world have bilingual lexicons or word lists available(Wang et al., 2022).

*Representation degeneration* is a prevalent problem in text generation (Gao et al., 2018) and machine translation models (Kudugunta et al., 2019). Contrastive learning (Hadsell et al., 2006) aims to bring similar sentences in the model close together and dissimilar sentences far from each other in the representation space. This is an effective solution to the representation problem in machine translation (Pan et al., 2021; Li et al., 2022). However, the naïve contrastive learning framework that utilizes random non-target sequences as negative examples is suboptimal, because they are easily distinguishable from the correct output (Lee et al., 2020).

To address both problems mentioned above, we present a novel multilingual NMT approach which leverages plentiful data sources: target-side monolingual data and a bilingual dictionary. Specifically, we start by using constrained beam search (Post and Vilar, 2018) to construct pseudo-parallel data in an online mode. To overcome the data imbalance problem, we propose training a bidirectional autoencoder, while to address representation degeneration, we use bidirectional contrastive learning. Finally, we use a curriculum learning (Bengio et al., 2009) sampling strategy. This uses the score given by token coverage in the bilingual dictionary to rearrange the order of training examples, such that sentences with more tokens in the dictionary are seen earlier and more frequently during training.

In summary, we make the following contributions: *i)* We propose a novel approach that uses only target-side monolingual data and a bilingual dictionary to improve MNT performance. *ii)* We define two modules, bidirectional autoencoder and bidirectional contrastive learning, to address the data imbalance and representation degeneration prob-

lem. *iii)* We show that our method demonstrates zero-shot domain transfer and language transfer capability. *iv)* We also show that our method is an effective solution for both the repetition (Fu et al., 2021) and the off-target (Zhang et al., 2020a) problems in large-scale MNMT models.

## 2    Related Work

**Multilingual Neural Machine Translation.** MNMT is rapidly moving towards developing large models that enable translation between an increasing number of language pairs. Fan et al. (2021) proposed *m2m_100* model that enables translation between 100 languages. Siddhant et al. (2022) and Costa-jussà et al. (2022) extend the current MNMT models to support translation between more than 200 languages using supervised and self-supervised learning methods.

**Autoencoder.** An autoencoder (AE) is a generative model that is able to generate its own input. There are many variants of AE that can be useful for machine translation. Zhang et al. (2016) and Eikema and Aziz (2019) propose using a variational autoencoder to improve the performance of machine translation models. A variant of the same generative model is the denoising autoencoder, which is an important component of unsupervised machine translation models (Lample et al., 2018a). However, the utility of autoencoders has not been fully explored for MNMT. To the best of our knowledge, we are the first to propose training an autoencoder using only target-side monolingual data and a bilingual dictionary to improve low-resource MNMT.

**Contrastive Learning.** Contrastive learning is a technique that clusters similar data together in a representation space while it simultaneously separates the representation of dissimilar sentences. It is useful for many natural language processing tasks (Zhang et al., 2022). Recently, Pan et al. (2021) and Vamvas and Sennrich (2021) used contrastive learning to improve machine translation and obtained promising results. However, these methods use the random replacing technique to construct the negative examples, which often leads to a significant divergence between the semantically similar sentences and the ground-truth sentence in the model representation space. This large changes makes the model more difficult to distinguish correct sentence from incorrect ones. We use small perturbations to construct negative examples, ensuring their proximity to the ground-truth sentence within the semantic

space, which significantly mitigates the aforementioned issue.

# 3 Method

Our goal is to overcome the *data imbalance* and *representation degeneration* issues in the MNMT model. We aim to improve the performance of MNMT without using parallel data, instead relying only on target-side monolingual data and a bilingual dictionary. Our approach contains three parts: online pseudo-parallel data construction (Section 3.1), bidirectional autoencoder (Section 3.2) and bidirectional contrastive learning (Section 3.3). Figure 1 illustrates the overview of our method. The architectures of the bidirectional autoencoder (left) and bidirectional contrastive learning (right) are presented in Figure 2.

## 3.1 Online Pseudo-Parallel Data Construction

Let us assume that we want to improve performance when translating from source language $\ell_s$ to target language $\ell_t$. We start with a monolingual set of sentences from the target language, denoted as $\mathcal{D}_{mono}^{\ell_t}$, a bilingual dictionary, denoted as $\mathcal{D}_{dict}^{\ell_t \to \ell_s}$, and a target monolingual sentence with $tt$ tokens, denoted as $X_i^{\ell_t} = \{x_1, ..., x_{tt}\}$, $X_i^{\ell_t} \in \mathcal{D}_{mono}^{\ell_t}$. We use lexically constrained decoding (i.e., constrained beam search; Post and Vilar, 2018) to generate a pseudo source language sentence $X_i^{\ell_s} = \{x_1, ..., x_{ss}\}$ in an online mode:

$$X_i^{\ell_s} = gen(X_i^{\ell_s} | \theta, X_i^{\ell_t}, \mathcal{D}_{dict}^{\ell_t \to \ell_s}) \quad (1)$$

where $gen(\cdot)$ is the lexically constrained beam search function and $\theta$ denotes the parameters of the model. It is worth noting that parameters $\theta$ will not be updated during the generation process, but will be updated in the following steps (Section 3.2 and Section 3.3).

## 3.2 Bidirectional Autoencoder

An autoencoder (Vincent et al., 2008) first aims to learn how to efficiently compress and encode data, then to reconstruct the data back from the reduced encoded representation to a representation that is as close as possible to the original input.

We propose performing autoencoding using only target-side monolingual data. This is different from prior work on UNMT, which uses both source and target-side data (Lample et al., 2018a). Our bidirectional autoencoder contains two parts: backward autoencoder (Section 3.2.1) and forward autoencoder (Section 3.2.2).

### 3.2.1 Backward Autoencoder

After we obtain $X_i^{\ell_s}$ from Eq. 1, we have the pseudo-parallel pairs $(X_i^{\ell_t}, X_i^{\ell_s}) \in \mathcal{D}_{pse}$. Then, we feed $X_i^{\ell_t}$ to the MNMT model to get the contextual output embedding $\mathbf{Z}_{\mathbf{bkd}}^{\ell_s}$. Formally, the encoder generates a contextual embedding $\mathbf{H}_i^{\ell_t}$ given $X_i^{\ell_t}$ and $l_t$ as input, which is in turn given as input to the decoder (together with $l_s$) to generate $\mathbf{Z}_{\mathbf{bkd}}^{\ell_s}$:

$$\begin{aligned} \mathbf{H}_{\mathbf{i}}^{\ell_{\mathbf{t}}} &= \text{Encoder}(X_i^{\ell_t}, \ell_t), \\ \mathbf{Z}_{\mathbf{bkd}}^{\ell_{\mathbf{s}}} &= \text{Decoder}(\mathbf{H}_{\mathbf{i}}^{\ell_{\mathbf{t}}}, \ell_s) \end{aligned} \quad (2)$$

Finally, the backward autoencoder loss is formulated as follows:

$$\mathcal{L}_{\text{AE\_bkd}} = -\sum \log P_\theta \left( X_i^{\ell_s} \mid X_i^{\ell_t}, \mathbf{Z}_{\mathbf{bkd}}^{\ell_{\mathbf{s}}} \right) \quad (3)$$

### 3.2.2 Forward Autoencoder

Given $\mathbf{Z}_{\mathbf{bkd}}^{\ell_{\mathbf{s}}}$ from Eq. 2, we feed it to the MNMT model and get the contextual output denoted as $\mathbf{Z}_{\mathbf{fwd}}^{\ell_{\mathbf{t}}}$:

$$\begin{aligned} \mathbf{H}_{\mathbf{i}}^{\ell_{\mathbf{s}}} &= \text{Encoder}(\mathbf{Z}_{\mathbf{bkd}}^{\ell_{\mathbf{s}}}, \ell_s), \\ \mathbf{Z}_{\mathbf{fwd}}^{\ell_{\mathbf{t}}} &= \text{Decoder}(\mathbf{H}_{\mathbf{i}}^{\ell_{\mathbf{s}}}, \ell_t) \end{aligned} \quad (4)$$

The forward auto-encoder loss is given by:

$$\mathcal{L}_{\text{AE\_fwd}} = -\sum \log P_\theta \left( X_i^{\ell_t} \mid \mathbf{Z}_{\mathbf{bkd}}^{\ell_{\mathbf{s}}}, \mathbf{Z}_{\mathbf{fwd}}^{\ell_{\mathbf{t}}} \right) \quad (5)$$

## 3.3 Bidirectional Contrastive Learning

The main challenge in contrastive learning is to construct the positive and negative examples. Naive contrastive learning (Pan et al., 2021) uses ground-truth target sentences as positive examples and random non-target sentences as negative examples. When a pretrained MNMT model is used, negative examples are initially located far from positive examples in the embedding space.

Motivated by Lee et al. (2020), we automatically generate negative and positive examples, such that both kinds of examples are difficult for the model to classify correctly. This arguably motivates learning meaningful representations. Different from Lee et al. (2020), we construct *negative* examples in the *source-side sentence*[3] and *positive* examples in the *target-side sentence*, instead of constructing

---

[3]The source-side sentence here denotes the *source-side* in *both directions* and is not a sentence in the source language. This is also the case for the target sentence.

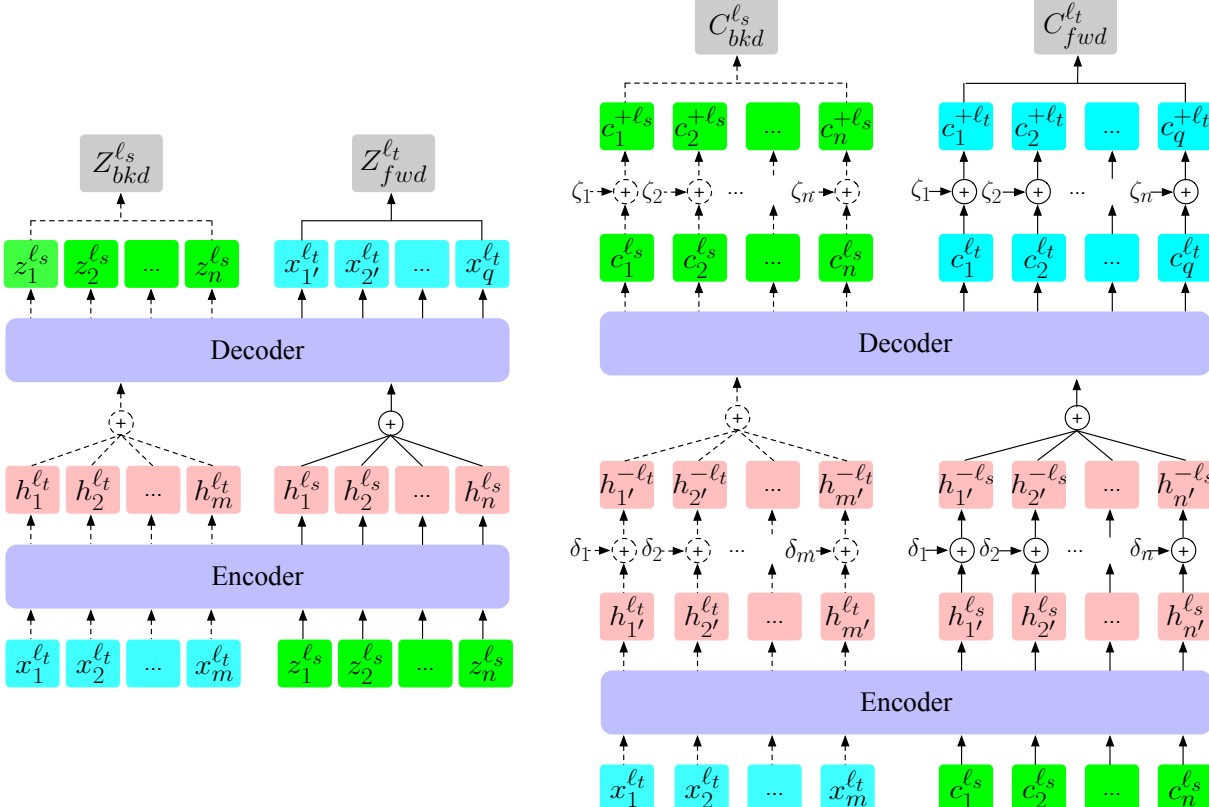

Figure 2: Model architecture: bidirectional autoencoder (left) and bidirectional contrastive learning (right). The symbols are consistent with the description in Section 3.

both in the *target-side sentence*. Specifically, to generate a negative example, we add a small perturbation $\delta_i = \{\delta_1 \dots \delta_T\}$ to $\mathbf{H_i}$, which is the hidden representation of the *source-side sentence*. We construct positive examples by adding a perturbation $\zeta_i = \{\zeta_1 \dots \zeta_T\}$ to $\mathbf{H_i}$, which is the hidden state of the *target-side sentence*. Different from Pan et al. (2021) who use a random replacing technique that may result in meaningless negative examples (already well-discriminated in the embedding space), we add the perturbation to ensure that the resulting embedding space is not already in a close proximity or far apart from the original embedding space. More details on how to generate the perturbation of $\delta_i$ and $\zeta_i$ can be found in Appendix A.

### 3.3.1 Backward Contrastive Learning

Given pseudo-parallel pairs $(X_{i'}^{\ell_t}, X_{i'}^{\ell_s}) \in \mathcal{D}_{pse}$ from Eq. 1, we first feed $X_{i'}^{\ell_t}$ to the MNMT model to generate the contextual embedding $\mathbf{H}_{i'}^{\ell_t}$. Then, we add a small perturbation $\delta_{bkd}^{(i')}$ after $\mathbf{H}_{i'}^{\ell_t}$ to form the negative example, denoted as $\mathbf{H}_{i'}^{-\ell_t}$. Finally, the contextual output of the decoder $\mathbf{C}_{\mathbf{bkd}}^{\ell_s}$ is generated by feeding $H_{i'}^{-\ell_t}$ to the decoder, and the positive ex-

ample $\mathbf{H}_{i'}^{+\ell_s}$ is generated by adding another small perturbation $\zeta_{bkd}^{(i')}$.

$$
\begin{aligned}
\mathbf{H}_{\mathbf{i'}}^{\ell_t} &= \text{Encoder}(X_{i'}^{\ell_t}, \ \ell_t), \\
\mathbf{H}_{\mathbf{i'}}^{-\ell_t} &= \mathbf{H}_{\mathbf{i'}}^{\ell_t} + \delta_{bkd}^{(i')}, \\
\mathbf{C}_{\mathbf{bkd}}^{\ell_s} &= \text{Decoder}(\mathbf{H}_{\mathbf{i'}}^{-\ell_t}, \ell_s), \\
\mathbf{H}_{\mathbf{i'}}^{+\ell_s} &= \text{Decoder}(\mathbf{C}_{\mathbf{bkd}}^{\ell_s}, \ell_s) + \zeta_{bkd}^{(i')},
\end{aligned}
\tag{6}
$$

Finally, the backward contrastive learning loss is formulated as follows:

$$
\mathcal{L}_{\text{CL\_bkd}} = -\sum \log \frac{e^{\text{sim}^+(\mathbf{H}_{\mathbf{i'}}^{\ell_t}, \mathbf{H}_{\mathbf{i'}}^{+\ell_s})/\tau}}{\sum_{\mathbf{H}_{\mathbf{i'}}^{-\ell_t}} e^{\text{sim}^-(\mathbf{H}_{\mathbf{i'}}^{\ell_t}, \mathbf{H}_{\mathbf{i'}}^{-\ell_t})/\tau}}
\tag{7}
$$

### 3.3.2 Forward Contrastive Learning

After we get $\mathbf{C}_{\mathbf{bkd}}^{\ell_s}$ from Eq. 6, we feed $\mathbf{C}_{\mathbf{bkd}}^{\ell_s}$ and the small perturbation $\delta_{\mathbf{fwd}}^{(\mathbf{i'})}$ to the MNMT model to obtain the contextual output denoted as $\mathbf{C}_{\mathbf{fwd}}^{\ell_t}$ and the negative example $\mathbf{H}_{i'}^{-\ell_s}$. Then, we feed $\mathbf{C}_{\mathbf{fwd}}^{\ell_t}$ and another small perturbation $\zeta_{fwd}^{(i')}$ to generate a positive example denoted as $\mathbf{H}_{\mathbf{i'}}^{+\ell_t}$.

$$\mathbf{H}_{\mathbf{i'}}^{\ell_{\mathbf{s}}} = \text{Encoder}(\mathbf{C}_{\mathbf{bkd}}^{\ell_{\mathbf{s}}}, \ell_s),$$

$$\mathbf{H}_{\mathbf{i'}}^{-\ell_{\mathbf{s}}} = \mathbf{H}_{\mathbf{i'}}^{\ell_{\mathbf{s}}} + \delta_{fwd}^{(i')},$$

$$\mathbf{C}_{\mathbf{fwd}}^{\ell_{\mathbf{t}}} = \text{Decoder}(\mathbf{H}_{\mathbf{i'}}^{-\ell_{\mathbf{s}}}, \ell_t), \quad (8)$$

$$\mathbf{H}_{\mathbf{i'}}^{+\ell_{\mathbf{t}}} = \text{Decoder}(\mathbf{C}_{\mathbf{fwd}}^{\ell_{\mathbf{t}}}, \ell_t) + \zeta_{fwd}^{(i')},$$

Finally, the forward contrastive learning loss is given by the following equation:

$$\mathcal{L}_{\text{CL\_fwd}} = -\sum \log \frac{e^{\text{sim}^+(\mathbf{H}_{\mathbf{i'}}^{\ell_{\mathbf{s}}}, \mathbf{H}_{\mathbf{i'}}^{+\ell_{\mathbf{t}}})/\tau}}{\sum_{\mathbf{H}_{\mathbf{i'}}^{-\ell_{\mathbf{s}}}} e^{\text{sim}^-(\mathbf{H}_{\mathbf{i'}}^{\ell_{\mathbf{s}}}, \mathbf{H}_{\mathbf{i'}}^{-\ell_{\mathbf{s}}})/\tau}} \quad (9)$$

### 3.4 Curriculum Learning

Curriculum Learning (Bengio et al., 2009) suggests starting with easier tasks and progressively gaining experience to process more complex tasks, which has been proved to be useful in machine translation (Stojanovski and Fraser, 2019; Zhang et al., 2019; Lai et al., 2022b). In our training process, we first compute token coverage for each monolingual sentence using the bilingual dictionary. This score is used to determine a curriculum to sample the sentences for each batch, so that higher-scored sentences are selected early on during training.

### 3.5 Training Objective

The model can be trained by minimizing a composite loss from Eq. 3, 5, 7 and 9 as follows:

$$\mathcal{L}^* = \lambda(\mathcal{L}_{AE\_bkd} + \mathcal{L}_{AE\_fwd}) + \\ (1 - \lambda)(\mathcal{L}_{CL\_bkd} + \mathcal{L}_{CL\_fwd}) \quad (10)$$

Where $\lambda$ is the balancing factor between the autoencoder and contrastive learning component.

## 4 Experiments

**Datasets.** We conduct three group of experiments: *bilingual setting*, *multilingual setting* and *high-resource setting*. In the *bilingual setting*, we focus on improving the performance on a specific long-tail language pair. We choose 10 language pairs at random that have BLEU < 2.5 in the original *m2m_100* model and a considerable amount of monolingual data in the target language in *news-crawl*.[4] The language pairs cover the following languages (ISO 639-1 language code[5]): *en, ta, kk,*

[4]https://data.statmt.org/news-crawl
[5]https://en.wikipedia.org/wiki/List_of_ISO_639-1_codes

*ar, ca, ga, bs, ko, ka, tr, af, hi, jv, ml*. In the *multilingual setting*, we aim to improve the performance on long-tail language pairs, which share the same target language. We randomly select 10 languages where the average BLEU score on the language pairs with the same target language is less than 2.5. For the languages not covered from *news-crawl*, we use the monolingual data from CCAligned[6] (El-Kishky et al., 2020). The languages we use are: *ta, hy, ka, be, kk, az, mn, gu*. For the *high-resource setting*, we aim to validate whether our proposed method also works for high-resource languages. We randomly select 6 language pairs that cover the following language codes: *en, de, fr, cs*.

**Dictionaries.** We extract bilingual dictionaries using the *wiktextract*[7] tool. For pairs not involving English, we pivot through English. Given a source language $\ell_s$ and a target language $\ell_t$, the intersection of the two respective bilingual dictionaries with English creates a bilingual dictionary $\mathcal{D}_{dict}^{\ell_s \to \ell_t}$ from $\ell_s$ to $\ell_t$. The statistics of the dictionaries can be seen in Appendix D.1.

**Data Preprocessing.** For the monolingual data, we first use a language detection tool[8] (*langid*) to filter out sentences with mixed language. We proceed to remove the sentences containing at least 50% punctuation and filter out duplicated sentences. To control the influence of corpus size on our experimental results, we limit the monolingual data of all languages to 1M. For dictionaries, we also use *langid* to filter out the wrong languages both on the source and target side.

**Baselines.** We compare our methods to the following baselines: i) **m2m**: Using the original *m2m_100* model (Fan et al., 2021) to generate translations. ii) **pivot_en**: Using English as a pivot language, we leverage *m2m_100* to translate target-side monolingual data to English and then translate English to the source language. Following this method, we finetune the *m2m_100* model using the pseudo-parallel data. iii) **BT**: Back-Translate (Sennrich et al., 2016) target-side monolingual data using *m2m_100* model to generate the pseudo source-target parallel dataset, then finetune the *m2m_100* model using this data. iv) **wbw_lm**: Use a bilingual dictionary, cross-lingual word embeddings and a target-side language model to translate word-by-word and then improve the translation through a

[6]https://opus.nlpl.eu/CCAligned.php
[7]https://github.com/tatuylonen/wiktextract
[8]https://fasttext.cc/docs/en/language-identification.html

| Models | Bilingual Setting | | | | | | | | | |
|---|---|---|---|---|---|---|---|---|---|---|
| | en→ta | en→kk | ar→ta | ca→ta | ga→bs | kk→ko | ka→ar | ta→tr | af→ta | hi→kk |
| m2m | 2.12 | 0.26 | 0.34 | 1.75 | 0.51 | 0.85 | 2.14 | 1.41 | 1.46 | 0.84 |
| pivot_en | - | - | 0.30 | 0.74 | 0.00 | 0.27 | 0.15 | 1.38 | 1.00 | 0.22 |
| BT | 0.76 | 0.67 | 0.60 | 1.13 | 0.63 | 0.97 | 0.06 | 2.05† | 0.72 | 0.43 |
| wbw_lm | 2.76† | 0.36 | 0.87 | 0.68 | 0.36 | 0.07 | 2.86† | 2.26† | 1.47 | 0.04 |
| syn_lexicon | 1.33 | 0.14 | 0.72 | 2.07† | 0.93 | 1.10† | 0.85 | 0.57 | 2.07† | 0.89 |
| **Bi-ACL** w/o Curriculum | 4.57‡ | 1.35‡ | 1.76‡ | 3.14‡ | 1.81‡ | 3.07‡ | 3.92‡ | 4.18‡ | 3.15‡ | 1.53‡ |
| **Bi-ACL** (*ours*) | **5.14**‡ | **2.59**‡ | **2.32**‡ | **3.50**‡ | **2.37**‡ | **3.61**‡ | **4.76**‡ | **4.97**‡ | **3.68**‡ | **2.47**‡ |
| Δ | +3.02 | +2.33 | +1.98 | +1.75 | +1.86 | +3.03 | +2.62 | +3.56 | +2.22 | +1.63 |
| | Multilingual Setting | | | | | | | | | |
| | ta | hy | ka | be | kk | az | mn | gu | my | ga |
| m2m | 1.46 | 1.69 | 0.52 | 1.95 | 0.67 | 2.32 | 1.12 | 0.26 | 0.24 | 0.09 |
| **Bi-ACL** | **2.54**‡ | **3.17**‡ | **2.38**‡ | **3.12**‡ | **1.44**‡ | **3.28**‡ | **1.95**‡ | **1.18**‡ | **1.94**‡ | **1.37**‡ |
| Δ | +1.08 | +1.48 | +1.86 | +1.17 | +0.77 | +0.96 | +0.83 | +0.92 | +1.70 | +1.28 |
| | Multilingual Setting (specific language pair) | | | | | | | | | |
| | en→ta* | ar→ta* | ca→ta* | af→ta* | el→ta | en→kk* | hi→kk* | fa→kk | jv→kk | ml→kk |
| m2m | 2.12 | 0.34 | 1.75 | 1.46 | 1.21 | 0.26 | 0.84 | 0.54 | 1.77 | 0.69 |
| **Bi-ACL** | **5.37**‡ | **2.81**‡ | **3.82**‡ | **4.16**‡ | **3.24**‡ | **2.94**‡ | **2.91**‡ | **2.87**‡ | **3.73**‡ | **3.29**‡ |
| Δ | +3.25 | +2.47 | +2.07 | +2.70 | +2.03 | +2.68 | +2.07 | +2.33 | +1.96 | +2.60 |
| Φ | +0.23 | +0.49 | +0.32 | +0.48 | - | +0.35 | +0.44 | - | - | - |

Table 1: **Main Results**: BLEU scores for low-resource language pairs in the bilingual, multilingual setting, and 10 randomly selected language pairs in the multilingual setting. Language pairs with * in the multilingual setting are covered by the bilingual setting. Δ denotes improvement over the original *m2m_100* model, while Φ shows improvement over the bilingual setting. Best results are shown in bold. † and ‡ denotes significant over original m2m_100 model at 0.05/0.01, evaluated by bootstrap resampling (Koehn, 2004).

target-side denoising model (Kim et al., 2018). v) **syn_lexicon**: Replace the words in the target monolingual sentence with the corresponding source language words in a bilingual dictionary and use the pseudo-parallel data to finetune the *m2m_100* model (Wang et al., 2022).

**Implementation.** We use *m2m*, released in the HuggingFace repository[9] (Wolf et al., 2020). For the *wbw_lm* baseline, monolingual word embeddings are directly obtained from the *fasttext* website[10] and cross-lingual embeddings are trained using a bilingual dictionary as a supervision signal. We set $\lambda = 0.7$ in all our experiments (the effect of different $\lambda$ can be find in Appendix E.5).

**Evaluation.** We measure case-sensitive detokenized BLEU and statistical significant testing as implemented in SacreBLEU[11] All results are computed on the *devtest* dataset of Flores101[12] (Goyal et al., 2022). To evaluate the isotropy[13] of the MNMT model, we adopt the $I_1$ and $I_2$

isotropy measures from Wang et al. (2019), with $I_1(\mathbf{W}) \in [0, 1]$ and $I_2(\mathbf{W}) \geq 0$, where $\mathbf{W}$ is the model matrix from the whole model parameter $\theta$. Larger $I_1(\mathbf{W})$ and smaller $I_2(\mathbf{W})$ indicate a more isotropic embedding space in the MNMT model. Please refer to Appendix B for more details on $I_1$ and $I_2$.

## 5 Results

Table 1 shows the main results on low-resource language pairs in a bilingual and multilingual setting. Table 3 shows results on high-resource language-pairs in a bilingual setting, while Table 2 presents an isotropic embedding space analysis for the bilingual setting.

**Low-Resource Language Pairs in a Bilingual Setting.** As shown in Table 1, the baselines perform poorly and several of them are worse than the original *m2m_100* model. This can be attributed to the fact that their performance depends on the translation quality in the direction of source language to English and English to target language (*pivot_en*), the quality in the reverse direction (*BT*), the quality of cross-lingual word-embeddings (*wbw_lm*) and the token coverage in bilingual dictionary (*syn_lexicon*). Our method outperforms the baselines across all language pairs, even when

[9]github.com/huggingface/transformers
[10]https://fasttext.cc/docs/en/crawl-vectors.html
[11]github.com/mjpost/sacrebleu
[12]github.com/facebookresearch/flores

[13]The representation in MNMT model is not uniformly distributed in all directions but instead occupying a narrow cone in the semantic space, we call this 'anisotropy'.

| | ar→ta | | | | ta→tr | | | | de→fr | | | |
|---|---|---|---|---|---|---|---|---|---|---|---|---|
| | Encoder | | Decoder | | Encoder | | Decoder | | Encoder | | Decoder | |
| | $I_1 \uparrow$ | $I_2 \downarrow$ | $I_1 \uparrow$ | $I_2 \downarrow$ | $I_1 \uparrow$ | $I_2 \downarrow$ | $I_1 \uparrow$ | $I_2 \downarrow$ | $I_1 \uparrow$ | $I_2 \downarrow$ | $I_1 \uparrow$ | $I_2 \downarrow$ |
| m2m | 0.042 | 20.017 | 0.012 | 26.639 | 0.036 | 20.408 | 0.006 | 26.901 | 0.058 | 16.521 | 0.016 | 24.695 |
| pivot_en | 0.034 | 22.852 | 0.008 | 24.472 | 0.019 | 22.889 | 0.007 | 25.977 | 0.056 | 16.843 | 0.016 | 24.763 |
| BT | 0.011 | 25.825 | 0.007 | 25.797 | 0.028 | 22.009 | 0.009 | 27.492 | 0.074 | 14.774 | 0.015 | 24.878 |
| wbw_lm | 0.023 | 23.485 | 0.015 | 24.746 | 0.038 | 19.389 | 0.010 | 26.320 | 0.037 | 19.099 | 0.015 | 24.935 |
| syn_lexicon | 0.059 | 17.513 | 0.015 | 25.694 | 0.028 | 20.640 | 0.013 | 26.475 | 0.020 | 23.859 | 0.014 | 24.137 |
| **Bi-ACL** w/o Curriculum | 0.074 | 16.174 | 0.017 | 24.176 | 0.039 | 19.139 | 0.018 | 24.712 | 0.078 | 14.165 | 0.017 | 24.128 |
| **Bi-ACL** (*ours*) | **0.086** | **15.714** | **0.020** | **23.251** | **0.043** | **18.672** | **0.021** | **22.716** | **0.086** | **13.666** | **0.017** | **24.067** |

Table 2: **Main Results**: Isotropic embedding space analysis in ar→ta and ta→tr translation task. The definitions of $I_1$ and $I_2$ can be found in Appendix B.

| Models | en→de | en→fr | en→cs | de→fr | de→cs | fr→cs |
|---|---|---|---|---|---|---|
| m2m | 22.79 | 32.50 | 21.65 | 28.53 | 20.73 | 20.30 |
| pivot_en | - | - | - | 11.68 | 7.09 | 6.60 |
| BT | 24.08 | 27.71 | 21.52 | 19.45 | 17.41 | 17.00 |
| wbw_lm | 7.52 | 11.53 | 9.04 | 8.38 | 9.44 | 10.15 |
| syn_lexicon | 5.35 | 12.56 | 10.90 | 5.90 | 8.39 | 8.89 |
| **Bi-ACL** w/o Curriculum | 25.17 | 35.52 | 23.91 | 29.43 | 22.71 | 22.04 |
| **Bi-ACL** (*ours*) | **27.76** | **37.84** | **25.89** | **30.66** | **23.80** | **23.56** |
| Δ | +4.97 | +5.34 | +4.24 | +2.13 | +3.07 | +3.26 |

Table 3: **Main Results**: BLEU scores for high-resource language pairs in the bilingual setting.

the performance of the language pair is poor in the original *m2m_100* model. In addition, using the curriculum learning sampling strategy further improves our model's performance.

**Low-Resource Language Pairs in a Multilingual Setting.** In the middle part of Table 1, we show the average BLEU scores of all language pairs with the same target language. Our approach consistently shows promising results across all languages. Based on the results shown in the lower part of the same Table, we notice that the BLEU scores obtained in the multilingual setting on a specific language pair outperform the scores obtained in the bilingual setting. For example, we get 3.68 BLEU points for af→ta in the bilingual setting, while we get 4.16 in the multilingual setting. This confirms our intuition that knowledge transfer between different languages in the MNMT model when using a multilingual setting is beneficial (see more details in Section 6.2).

**High-Resource Language Pairs in a Bilingual Setting.** As shown in Table 3, baseline systems do not perform well on all high-resource pairs due to the same reasons as in the long-tail languages setting. Our approach outperforms the baselines on all high-resource pairs. In addition, curriculum learning takes full advantage of the original model in the high-resource setting, with stronger gains in performance than in the low-resource setting. In-

terestingly, our findings reveal that back translation does not yield optimal results in both low and high resource settings. In low-resource languages, the performance of the language pair and its reverse direction in the original m2m_100 model is significantly poor (i.e., nearly zero). Consequently, the use of back-translation results in a performance that is inferior to that of m2m_100. For high-resource languages, the language pairs already exhibit strong performance in the original m2m_100 model. This makes it challenging to demonstrate that the incorporation of additional pseudo-parallel data can outperform the non-utilization of the pseudo-corpus. Another potential concern is that the large amount of monolingual data we employ, coupled with the substantial amount of pseudo-parallel data derived from back translation, may disrupt the pre-trained model. This observation aligns with the findings of Liao et al. (2021) and Lai et al. (2021).

**Statistical Significance Tests.** The use of BLEU in isolation as the single metric for evaluating the quality of a method has recently received criticism (Kocmi et al., 2021). Therefore, we conduct statistical significance testing in the low-resource setting to demonstrate the difference as well as the superiority of our method over other baseline systems. As can be seen in Table 1, our method outperforms the baseline by significant differences, which is even more evident in the case study in Table 10. This is because the baseline system faces the serious problems of generating duplicate words (repeat problem) and translating to the wrong language (off-target problem), while our method avoids these two problems.

**Isotropy Analysis.** It is clear from Table 2 that the embedding space on the encoder side is more isotropic than on the decoder side. This is because we only use the target-side monolingual data to

| # | $\mathcal{L}_{AE\_bkd}$ | $\mathcal{L}_{AE\_fwd}$ | $\mathcal{L}_{CL\_bkd}$ | $\mathcal{L}_{CL\_fwd}$ | en→ta | | | ta→tr | | | en→de | | |
|---|---|---|---|---|---|---|---|---|---|---|---|---|---|
| | | | | | BLEU | $I_1 \uparrow$ | $I_2 \downarrow$ | BLEU | $I_1 \uparrow$ | $I_2 \downarrow$ | BLEU | $I_1 \uparrow$ | $I_2 \downarrow$ |
| #1 | √ | × | × | × | 2.51 | 0.005 | 30.737 | 3.34 | 0.004 | 32.378 | 23.14 | 0.011 | 24.876 |
| | × | √ | × | × | **3.27** | 0.006 | 29.299 | **3.96** | 0.006 | 29.373 | **23.82** | 0.011 | 24.651 |
| | × | × | √ | × | 2.39 | 0.008 | 26.562 | 2.69 | 0.007 | 28.663 | 22.57 | 0.013 | 24.872 |
| | × | × | × | √ | 2.36 | **0.009** | **26.541** | 2.65 | **0.007** | **27.155** | 22.89 | **0.013** | **24.367** |
| #2 | √ | √ | × | × | **4.03** | 0.009 | 27.147 | **4.12** | 0.011 | 27.039 | **25.62** | 0.014 | 24.075 |
| | √ | × | √ | × | 2.36 | 0.012 | 26.782 | 3.64 | 0.010 | 27.636 | 24.84 | 0.013 | 24.513 |
| | √ | × | × | √ | 2.50 | 0.014 | 26.007 | 3.37 | 0.012 | 26.881 | 24.36 | 0.012 | 24.841 |
| | × | √ | √ | × | 3.54 | 0.012 | 26.964 | 3.89 | 0.011 | 27.175 | 24.59 | 0.012 | 24.764 |
| | × | √ | × | √ | 3.81 | **0.019** | **25.597** | 4.03 | **0.015** | **26.460** | 25.17 | **0.014** | **24.025** |
| | × | × | √ | √ | 2.53 | 0.013 | 28.459 | 3.61 | 0.012 | 27.639 | 24.73 | 0.012 | 24.723 |
| #3 | √ | √ | √ | × | 3.85 | 0.020 | 24.732 | 3.73 | 0.016 | 26.197 | 25.43 | 0.014 | 24.137 |
| | √ | √ | × | √ | **4.31** | **0.028** | **23.861** | **4.29** | **0.019** | **25.573** | **26.44** | **0.015** | **24.019** |
| | √ | × | √ | √ | 2.82 | 0.023 | 24.352 | 3.77 | 0.018 | 25.852 | 25.63 | 0.014 | 24.257 |
| | × | √ | √ | √ | 3.83 | 0.025 | 24.173 | 4.05 | 0.015 | 26.447 | 26.17 | 0.015 | 14.192 |
| #4 | √ | √ | √ | √ | **5.14** | **0.031** | **22.392** | **4.97** | **0.022** | **24.175** | **27.76** | **0.016** | **23.951** |

Table 4: Ablation study of four loss functions on en→ta and ta→ar translation task. "√" means the loss function is included in the training objective while "×" means it is not. Both $I_1$ and $I_2$ score are computed in the decoder side.

improve the decoder of the MNMT model. Compared to other baseline systems, we get a higher $I_1$ and lower $I_2$ score, which shows a more isotropic embedding space in our methods. An interesting finding is that the difference in isotropic space between high-resource language pairs is not significant. This phenomenon is because the original m2m_100 model already performs very well on high-resource language pairs and the representation degeneration is not substantial for those language pairs. In addition, the phenomenon is consistent with the findings in Table 4.

# 6 Analysis

In this section, we conduct additional experiments to better understand the strengths of our proposed methods. We first investigate the impact of four components on the results through an ablation study (Section 6.1). Then, we evaluate the zero-shot domain transfer ability and language transfer ability of our method (Section 6.2). Finally, we evaluate some impact factors (the quality of bilingual dictionary and the amount of monolingual data) on our proposed method (Section 6.3) and present a case study to show the strengths of our approach in solving the repetition problem and off-target issues in MNMT model.

## 6.1 Ablation Study

Our training objective function, shown in Eq. 10, contains four loss functions. We perform an ablation study on *en→ta*, *ta→ar* and *en→de* translation tasks to understand the contribution of each loss function. The experiments in Table 4 are divided into four groups, each group representing the number of loss functions. We have the following three findings: i) #1 clearly shows that the bidirectional autoencoder losses ($\mathcal{L}_{AE\_bkd}$ and $\mathcal{L}_{AE\_fwd}$) play a more critical role than the bidirectional contrastive learning losses ($\mathcal{L}_{CL\_bkd}$ and $\mathcal{L}_{CL\_fwd}$) in terms of BLEU score. However, bidirectional contrastive losses are more important than bidirectional autoencoder losses in terms of $I_1$ and $I_2$ score. This could be the case because contrastive learning aims to improve the MNMT model's isotropic embedding space rather than the translation from source language to target language. ii) Using forward direction losses results in a better translation quality compared to backward direction losses (#1). This is because our goal is to improve the performance from source language to target language, which is the forward direction in the loss functions. iii) The more loss functions there are, the better the performance. The combination of all four loss functions yields the best performance. iv) We show that the $I_1$ and $I_2$ scores in high-resource language pairs (en→de) do not have a significant change as the original embedding space is already isotropic.

## 6.2 Domain Transfer and Language Transfer

Motivated by recent work on the domain and language transfer ability of MNMT models (Lai et al., 2022a), we conduct a number of experiments with extensive analysis to validate the zero-shot domain

transfer ability, as well as the language transfer ability of our proposed method. We have the following findings: i) Our proposed method works well not only on the Flores101 datasets (domains similar to training data of the original *m2m_100* model), but also on other domains. This supports the domain transfer ability of our proposed method. ii) We show that the transfer ability is more obvious in the multilingual setting than in the bilingual setting, which is consistent with the conclusion from Table 6 in the multilingual setting. More details can be found in Appendix E.3 and E.4.

### 6.3 Further Investigation

To investigate two other important factors in our proposed methods, we conducted additional experiments to evaluate the impact of the quality of the dictionary and the amount of monolingual data. In general, we observe that better performance can be obtained by utilizing a high-quality bilingual dictionary. In addition, the size of the monolingual data used is not proportional to the performance improvement. More details can be found in Appendix E.1 and E.2. Also, compared with the baseline models, our method has strengths in solving repetition and off-target problems, which are two common issues in large-scale MNMT models. More details can be found in Appendix E.6.

## 7 Conclusion

To address the data imbalance and representation degeneration problem in MNMT, we present a framework named *Bi-ACL* which improves the performance of MNMT models using only target-side monolingual data and a bilingual dictionary. We employ a bidirectional autoencoder and bidirectional contrastive learning, which prove to be effective both on long-tail languages and high-resource languages. We also find that *Bi-ACL* shows language transfer and domain transfer ability in zero-shot scenarios. In addition, *Bi-ACL* provides a paradigm that an inexpensive bilingual lexicon and monolingual data should be fully exploited when there are no bilingual parallel corpora, which we believe more researchers in the community should be aware of.

## 8 Limitations

This work has two main limitations. i) We only evaluated the proposed method on the machine translation task, however, *Bi-ACL* should work well on other NLP tasks, such as text generation or question answering task, because our framework only depends on the bilingual dictionary and monolingual data, which can be easily found on the internet for many language pairs. ii) We only evaluated *Bi-ACL* using *m2m_100* as a pretrained model. However, we believe that our approach would also work with other pretrained models, such as mT5 (Xue et al., 2021) and mBART (Liu et al., 2020). Because the two components (bidirectional autoencoder and bidirectional contrastive learning) we proposed can be seen as plugins, they could be easily added to any pretrained model.

## Acknowledgement

This work was supported by funding from China Scholarship Council (CSC). This work has received funding from the European Research Council under the European Union's Horizon 2020 research and innovation program (grant agreement #640550). This work was also supported by the DFG (grant FR 2829/4-1).

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

## A Contrastive Learning

Our approach is different from traditional contrastive learning, which takes a ground-truth sentence pair as a positive example and a random nontarget sentence pair in the same batch as a negative example. Motivated by Lee et al. (2020), we construct positive and negative examples automatically.

### A.1 Negative Example Formulation

As described in Section 3.3, to generate a negative example, we add a small perturbation $\delta_i = \{\delta_1 \ldots \delta_T\}$ to the $\mathbf{H_i}$, which is the hidden representation of the *source-side sentence*. As seen in Eq. 6, the negative example is denoted as $\mathbf{H_{i'}^{-\ell_t}}$, and is formulated as the sum of the original contextual embedding $\mathbf{H_{i'}^{\ell_t}}$ of the target language sentence $X_{i'}^{\ell_t}$ and the perturbation $\delta_{bkd}^{(i')}$. Finally, $\delta_{bkd}^{(i')}$ is formulated as the conditional log likelihood with respect to $\delta$. $\delta_{bkd}^{(i')}$ is semantically very dissimilar to $X_{i'}^{\ell_t}$, but very close to the hidden representation $\mathbf{H_{i'}^{-\ell_t}}$ in the embedding space.

$$\mathbf{H_{i'}^{\ell_t}} = \text{Encoder}(X_{i'}^{\ell_t},\ \ell_t),$$
$$\mathbf{H_{i'}^{-\ell_t}} = \mathbf{H_{i'}^{\ell_t}} + \delta_{bkd}^{(i')}$$
$$\delta_{bkd}^{(i')} = \underset{\delta, \|\delta\|_2 \leq \epsilon}{\arg\min} \log p_\theta \left( X_{i'}^{\ell_s} \mid X_{i'}^{\ell_t}; \mathbf{H_{i'}^{\ell_t}} + \delta \right)$$

where $\epsilon \in (0, 1]$ is a parameter that controls the perturbation and $\theta$ denotes the parameters of the MNMT model.

### A.2 Positive Example Formulation

As shown in Eq. 6, we create a positive example of the target sentence by adding a perturbation $\zeta_i = \{\zeta_1 \ldots \zeta_T\}$ to $\mathbf{H_i}$, which is the hidden state of the *target-side sentence*. The objective of the perturbation $\zeta_{bkd}^{(i')}$ is to minimize the KL divergence between the perturbed conditional distribution and the original conditional distribution as follows:

$$\zeta_{bkd}^{(i')} = \arg\min D_{KL} \left( p_{\theta^*} \left( X_{i'}^{\ell_s} \mid X_{i'}^{\ell_t} \right) \| p_\theta \left( \hat{X}_{i'}^{\ell_s} \mid \hat{X}_{i'}^{\ell_t} \right) \right), \tag{11}$$

where $\theta^*$ is the copy of the model parameter $\theta$. As a result, the positive example is semantically similar to $X_{i'}^{\ell_s}$ and dissimilar to the contextual embedding of the target sentence in the embedding space.

## B Evaluation of Isotropy

We use $I_1$ and $I_2$ scores from Wang et al. (2019) to characterize the isotropy of the output embedding

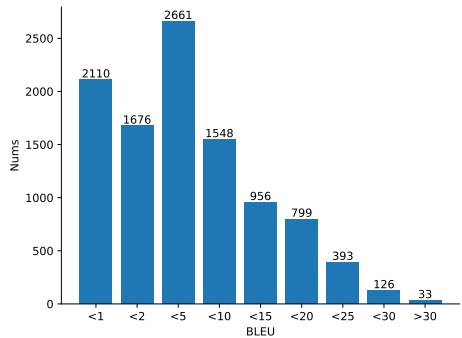

Figure 3: BLEU score statistics of the *m2m_100* model (Fan et al., 2021) on Flores101 dataset (Goyal et al., 2022) for $102 \times 101 = 10302$ language pairs. Each bar denotes the number of language pairs in the interval of the BLEU score.

space.

$$I_1(\boldsymbol{W}) = \frac{\min_{\boldsymbol{s} \in \boldsymbol{S}} Z(\boldsymbol{s})}{\max_{\boldsymbol{s} \in \boldsymbol{S}} Z(\boldsymbol{s})}$$
$$I_2(\boldsymbol{W}) = \sqrt{\frac{\sum_{\boldsymbol{s} \in \boldsymbol{S}} (Z(\boldsymbol{s}) - \bar{Z}(\boldsymbol{s}))^2}{|S| \bar{Z}(\boldsymbol{s})^2}}$$

where $Z(\boldsymbol{s}) = \sum_{i=1}^{n} \exp\left(\boldsymbol{s}^T \boldsymbol{w}_i\right)$ is close to some constant with high probability for all unit vectors $\boldsymbol{s}$ if the embedding matrix $\boldsymbol{W}$ is isotropic ($\boldsymbol{w_i} \in \boldsymbol{W}$). $\boldsymbol{S}$ is the set of eigenvectors of $\mathbf{W}^\top \mathbf{W}$. $I_2$ is the sample standard deviation of $Z(\boldsymbol{s})$ normalized by its average $\bar{Z}(\boldsymbol{s})$). We have $I_1(\boldsymbol{W}) \in [0, 1]$ and $I_2(\boldsymbol{W}) \geq 0$. Larger $I_1(\boldsymbol{W})$ and smaller $I_2(\boldsymbol{W})$ indicate more isotropic for word embedding space.

In this work, we randomly select 128 sentences from Flores101 benchmark to compute these two criteria. The results are shown in Table 2.

## C Model Configuration

We use the *m2m_100* model with 418MB parameters implemented in Huggingface. In our experiments, we use the AdamW (Loshchilov and Hutter, 2018) optimizer and the learning rate are initial to $2e-5$ with a dropout probability 0.1. We trained our models on one machine with 4 NVIDIA V100 GPUs. The batch size is set to 8 per GPU during training. To have a fair comparison, all experiments are trained for 3 epochs.

## D Statistics

### D.1 Statistics of BLEU scores in m2m_100

Figure 3 shows the BLEU scores of the m2m_100 model on all 10302 supported language pairs. We

| | en→ta | | | | ka→ar | | | | ta→tr | | | |
|---|---|---|---|---|---|---|---|---|---|---|---|---|
| | Flores | TED | QED | KDE | Flores | TED | QED | KDE | Flores | TED | QED | KDE |
| m2m | 2.12 | 2.19 | 1.16 | 0.57 | 2.14 | 4.47 | 1.00 | 13.35 | 1.41 | 2.03 | 1.32 | 8.10 |
| pivot_en | - | - | - | - | 0.15 | 0.21 | 0.08 | 0.53 | 1.38 | 1.34 | 0.86 | 3.93 |
| BT | 0.76 | 0.26 | 0.01 | 0.36 | 0.06 | 0.24 | 0.09 | 0.64 | 2.05 | 0.89 | 0.63 | 2.81 |
| wbw_lm | 2.76 | 1.97 | 0.29 | 0.67 | 2.86 | 0.17 | 0.12 | 6.24 | 2.26 | 1.53 | 1.19 | 8.70 |
| syn_lexicon | 1.33 | 1.19 | 0.08 | 0.20 | 0.85 | 1.65 | 0.45 | 10.64 | 0.57 | 0.41 | 0.55 | 7.08 |
| **Bi-ACL** w/o Curriculum | 4.57 | 2.62 | 1.24 | 0.75 | 3.92 | 4.61 | 1.20 | 13.82 | 4.18 | 2.39 | 1.68 | 11.75 |
| **Bi-ACL** | **5.14** | **2.84** | **1.41** | **1.05** | **4.76** | **4.93** | **1.57** | **14.62** | **4.97** | **2.81** | **1.96** | **12.74** |
| Δ | +3.02 | +0.65 | +0.25 | +0.48 | +2.62 | +0.46 | +0.57 | +1.27 | +3.56 | +0.78 | +0.64 | +4.64 |

Table 5: Domain transfer: BLEU scores on en→ta, ka→ar, ta→tr in different domains.

| | Bilingual Setting | |
|---|---|---|
| | en2ta⇒ar2ta | ar2ta⇒en2ta |
| m2m | 0.34 | 2.12 |
| pivot_en | - | 0.34 |
| BT | 0.28 | 0.55 |
| wbw_lm | 1.07 | 2.68 |
| syn_lexicon | 0.43 | 1.86 |
| **Bi-ACL** w/o curriculum | 2.08 | 4.53 |
| **Bi-ACL** transfer | **2.74** | **5.78** |
| Φ | +0.42 | +0.64 |
| | **Multilingual Setting** | |
| | ta⇒be | be⇒ta |
| m2m | 1.95 | 1.46 |
| **Bi-ACL** | 2.73 | 2.31 |
| **Bi-ACL** transfer | **3.67** | **3.42** |
| Φ | +0.55 | +0.88 |

Table 6: Language transfer: BLEU scores on langauge (pair) transfer ablity both on bilingual setting and multilingual setting. 'A⇒B' means from language (pair) A transfer to language (pair) B. Φ denotes the improvement on our proposed methods.

| Language Pair | #Size | Language Pair | #Size |
|---|---|---|---|
| en→ta | 8,376 | af→ta | 5,268 |
| en→kk | 11,323 | hi→kk | 24,762 |
| ar→ta | 26,768 | en→de | 68,029 |
| ca→ta | 18,757 | en→fr | 78,837 |
| ga→bs | 125,336 | en→cs | 35,879 |
| kk→ko | 38,710 | de→fr | 207,831 |
| ka→ar | 25,825 | de→cs | 125,909 |
| ta→tr | 14,169 | fr→cs | 104,510 |

Table 7: Statistics of bilingual dictionaries.

| | en→ta | ca→ta | ga→bs | ta→tr |
|---|---|---|---|---|
| m2m | 2.12 | 1.75 | 0.51 | 1.41 |
| panlex | 3.19 | 2.47 | 0.67 | 3.41 |
| wiktionary | **5.14** | **3.50** | **2.37** | **4.97** |

Table 8: The effect of bilingual dictionary quality on experimental performance in terms of BLEU score.

see that 21% of the langauge pairs have a BLEU score of almost 0 and more than 50% have a BLEU score of less than 5.

## D.2 Statistics of Bilingual Dictionaries

Table 7 shows the size of the bilingual dictionaries used in a bilingual setting. For the multilingual setting, we will publish our code to generate the bilingual dictionary for any language pair.

## E Further Analysis

### E.1 Quality of the Bilingual Dictionary

To investigate whether the quality of bilingual dictionary affects the performance of our method, we conduct additional experiments using the Panlex dictionary[14], a big dataset that covers 5,700 lan-

[14] https://panlex.org

guages. We evaluate the performance on en→ta, ca→ta, ga→bs and ta→tr translation tasks.

As seen in Table 8, using the dictionary mined from wikitionary results in a better performance than using the panlex dictionary. The reason for this is that, while Panlex supports bilingual dictionaries for many language pairs, we discovered that the quality of them is quite low, especially when English is not one of the two languages in the language pair.

### E.2 Amount of Monolingual Data

As described in Section 3.4, we use the bilingual dictionary coverage $\phi$ as the curriculum to order the training batch. In this section, we aim to investigate how the number of monolingual data affects the experimental results. A smaller $\phi$ means a larger number of monolingual data. We conduct experiments on en→ta, en→kk, ar→ta and ca→ta translation tasks with a different $\phi$.

As seen in Table 9, we observe that the amount

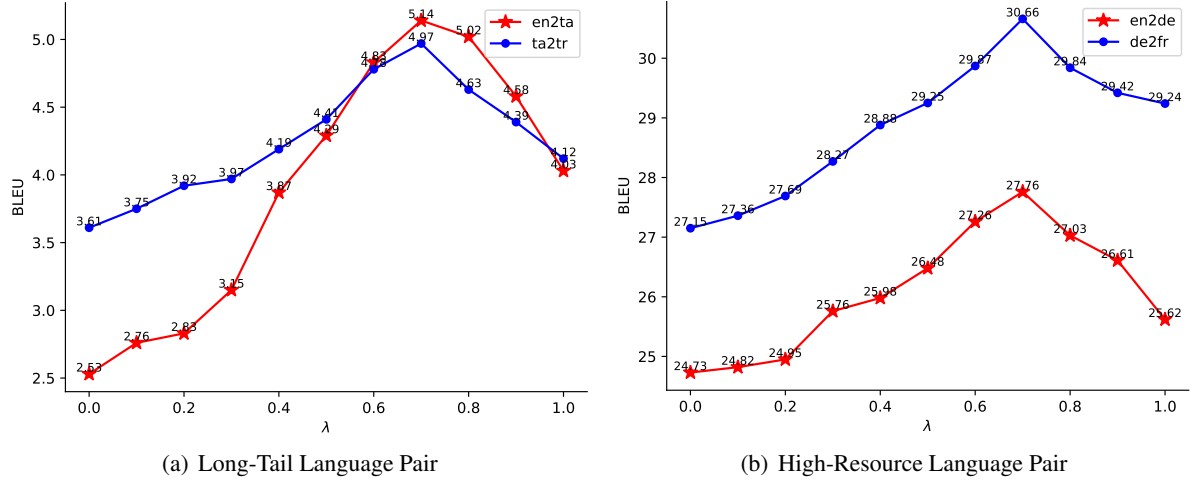

| | (a) Long-Tail Language Pair | (b) High-Resource Language Pair |
| --- | --- | --- |

Figure 4: The effect of different $\lambda$

|  | en→ta | en→kk | ar→ta | ca→ta |
| --- | --- | --- | --- | --- |
| m2m | 2.12 | 0.26 | 0.34 | 1.75 |
| $\phi = 0.5$ | 3.70 | 0.84 | 1.22 | 2.28 |
| $\phi = 0.6$ | 3.60 | 0.75 | 1.07 | 2.26 |
| $\phi = 0.7$ | 3.64 | 0.23 | 1.15 | 2.24 |
| $\phi = 0.8$ | 3.82 | 0.46 | 1.28 | 2.52 |
| $\phi = 0.9$ | **5.14** | **2.59** | **2.32** | **3.50** |
| $\phi = 1.0$ | 3.27 | 0.07 | 1.01 | 2.15 |

Table 9: The effect of monolingual corpus size in experimental results in terms of BLEU score. The smaller the value of $\phi$ (bilingual dictionary coverage) the larger the monolingual corpus.

of monolingual data is not proportional to the experimental performance. This is because a large percentage of words in a sentence are not covered by lexicons in the bilingual dictionaries, the performance of constrained beam search is limited. This phenomenon is consistent with the conclusion that the effect of the size of the pseudo parallel corpus in data augmentation (Fadaee et al., 2017) and back-translation (Sennrich et al., 2016) on the experimental results, i.e., that the performance of machine translation is not proportional to the size of the pseudo parallel corpus.

### E.3  Domain Transfer

To investigate the domain transfer ability of our approach, we first conduct experiments on en→ta, ka→ar, ta→tr translation tasks, then evaluate the performance in a zero-shot setting on three different domains (*TED*, *QED* and *KDE*) which are

publicly available datasets from OPUS[15] (Tiedemann, 2012) and on the Flores101 benchmark. The results is shown in Table 5.

According to Table 5, the performance of the baseline systems is even worse than the original *m2m_100* model, which suggests that they do not show domain robustness nor domain transfer ability due to poor performance (see Table 1). In contrast, our proposed method works well not only on the Flores101 datasets (domains similar to training data of the original *m2m_100* model), but also on other domains.

### E.4  Language Transfer

To investigate the language transfer ability, we use the model trained on a specific language (pair) to generate text for another language (pair) both in the bilingual and multilingual settings. For the bilingual setting, we run experiments to assess the language transfer ability between en→ta and ar→ta translation tasks. For the multilingual setting, we focus on translation scores between *ta* and *be*. The results is shown in Table 6.

As indicated in Table 6, we observe that the performance in our method outperforms the other baselines both in the bilingual and in the multilingual setting. We also discover that the transfer ability is more obvious in the multilingual setting than in the bilingual setting. This phenomenon is consistent with the conclusion from Table 1 in the multilingual setting. We believe that this can be attributed to the fact that in a multilingual setting, the language is used for all language pairs that share

---
[15]https://opus.nlpl.eu

| | Example 1 (Repetition Problem) |
|---|---|
| | **Source (English):** "We now have 4-month-old mice that are non-diabetic that used to be diabetic," he added. |
| | **Reference (Kazakh):** «Қазір бізде диабетпен ауырған, бірақ қазір диабеті жоқ 4 айлық тышқандар бар», - деп қосты ол. |
| m2m | «Жеңіске үлес қосқандар», «Жеңіске үлес қосқандар», «Жеңіске үлес қосқандар», «Жеңіске үлес қосқандар». |
| pivot_en | " Біз қазір Жеңіске бізге айлық Жеңіске келетін диабетті бізге пайдаланылып Жеңіске диабеттік," деді қосады. |
| BT | Сондай-ақ, бүгінгі таңда Жеңіске пайдаланатын 4 Жеңіске мешіт не диабет емес, деп Жеңіске ҚазАқпарат. |
| wbw_lm | " Біз қазір екенін айлық Мұқағали бізге келетін диабетті бізге пайдаланылып жатқанымыз диабеттік," деді қосады. |
| syn_lexicon | " біз қазір болу 4- ай бұрынғы мысал сол болу емес диабетик сол қосық болу диабетик", ол қосу. |
| Bi-ACL w/o Curriculum | Біз бізде диабетпен қосқандар, сол қазір диабеті айлық  4 айлық емес бар , ҚазАқпарат. |
| Bi-ACL (ours) | Біз бізде диабетпен қосқандар, сол қазір келетін айлық жоқ 4 айлық емес бар , ҚазАқпарат. |
| | Example 2 (Off-target Problem) |
| | **Source (English):** 'Their thermal behavior is not as steady as large caves on Earth that often maintain a fairly constant temperature, but it is consistent with these being deep holes in the ground," said Glen Cushing of the United States Geological Survey (USGS) Astrogeology Team and of Northern Arizona University located in Flagstaff, Arizona.' |
| | **Reference (Tamil):** அதிகாரிகள் வாக்காளரின் அடையாளத்தைச் சரி பார்த்த பிறகு, வாக்காளர் அந்த வாக்குப் பெட்டியில் உறையைப் போடுவார் மற்றும் வாக்காளர் பதிவேட்டில் கையொப்பம் இடுவார். பிரெஞ்சு தேர்தல் சட்டம், நடவடிக்கைகளை கடுமையாக செயல்படுத்துகிறது. |
| m2m | ஆங்கிலத்தில் இதை Single Orgasm, Multiple Orgasm என்றும் கூறுகிறார்கள். |
| pivot_en | பெரும்பாலும் இன்ஸ்டாகிராமில் உள்ள இந்த Glen Cushing of the United States Geological Survey (USGS) Astrogeology Team and of Northern Arizona University located in Flagstaff, Arizona. |
| BT | Their thermal behavior is not as steady as large caves on பார்த்த பிறகு, வாக்காளர் அந்த வாக்குப் அந்த வாக்குப் பெட்டியில் உறையைப் Glen Cushing of the United States Geological Survey (USGS) Astrogeology Team and of Northern Arizona University located in Flagstaff, Arizona. |
| wbw_lm | Their thermal behavior is not as steady as large caves அடையாளம், அந்த வாக்காளர் விழும் அந்த but it is consistent with these being deep holes in the ground, பிரான்ஸ் தேர்தல் சட்டம் போல் கட்டுப்பாடுகளுடன் நடைமுறைப்படுத்தும் இந்த வழக்கு. |
| syn_lexicon | Their thermal behavior is not as steady as large caves on Earth that அடிக்கடி பராமரிப்பு ஒரு மிக நிரந்தர வெப்பநிலை, ஆனால் அது ஒன்றாக கூட இந்த இருள் holes in the ground," என் Glen nited States Geological Survey (USGS) Astrogeology Team and of Northern Arizona University located in Flagstaff, Arizona. |
| Bi-ACL w/o Curriculum | அதிகாரிகள் வாக்காளரின் வாக்காளர் சரி பார்த்த பிறகு, வாக்காளர் அந்த வாக்குப் பௌர் விழும் அந்த வட்டா டுவார் மற்றும் வாக்காளர் பதிவேட்டில் அந்த வாக்கு. பிரெஞ்சு தேர்தல் சட்டம், நடவடிக்கைகளை கடுமையாக செயல்படுத்துகிறது. |
| Bi-ACL (ours) | அதிகாரிகள் அந்த வாக்காளர் அடையாளம், அந்த பிறகு, வாக்காளர் அந்த வாக்குப் பெட்டியில் உறையைப் போடுவார் மற்றும் வாக்காளர் பதிவேட்டில் மற்றும் கையெழுத்து அந்த வாக்கு சட்டம், நடவடிக்கைகளை முறைப்படுத்தும் இந்த வழக்கு. |

Table 10: Case study

the same target language, which can be seen as common information for all language pairs.

### E.5 The effect of $\lambda$

In Section 3.5, we set a $\lambda$ to balance the importance of both autoencoding and contrastive loss to our model. From Figure 4, we show that the autoencoding loss plays a more important role than contrastive loss in terms of BLEU. When $\lambda = 0.7$, we got the best performance both in long-tail language pair and high-resource langauge pair.

### E.6 Case Study

We now present qualitative results on how our method addresses the repetition and off-target problems. For the first example in Figure 10, we find that other baseline systems suffer from a severe repetition problem. This is attributed to a poor decoder.

In contrast, our method does not have a repetition problem, most likely because we enhanced the representation of the decoder through a bidirectional contrastive loss. For the second example, we show that while the off-target problem is prevalent in baseline systems, our method seems to provide an effective solution to it.