# OpenReview forum: "Mitigating Data Imbalance and Representation Degeneration in Multilingual Machine Translation"
_EMNLP/2023/Conference — EMNLP 2023 Findings_

### Official Review · Reviewer_G5qY · 2023-07-30

**Typos Grammar Style And Presentation Improvements:** 1. The related work section (section …
**Soundness:** 2

**Excitement:**

3: Ambivalent: It has merits (e.g., it reports state-of-the-art results, the idea is nice), but there are key weaknesses (e.g., it describes incremental work), and it can significantly benefit from another round of revision. However, I won't object to accepting it if my co-reviewers champion it.

**Paper Topic And Main Contributions:**

This paper proposes to mitigate two problems in multilingual machine translation: data imbalance and representation degeneration. To this end, they propose a Bi-ACL framework that contains a bidirectional autoencoder to make use of monolingual data as well as a bidirectional contrastive learning technique to help the model learn hidden representations better.

The contributions of this paper are mainly made toward the NLP engineering experiment direction. Their Bi-ACL framework demonstrates improved translation performance for low-resource languages in both bilingual and multilingual settings.

**Questions For The Authors:**

Question A: In line 272, it says H^{+l_t} while in the last formula of equation 6, it is H^{+l_s} instead. Meanwhile, no h^{+l_s} is in Figure 2 (right part). By h^{+l_s}, do you mean c^{+l_s} in the figure?  I also don't understand the Encoder part in the last formula in equation 6 because in the figure c^{+l_s} = c^{l_s}+perturbation and no Encoder is involved. I have similar confusion about the forward contrastive learning part.

Question B: Probably because of my confusion with equation 6, I also don't understand the logic behind your positive example construction. In lines 267 to 273, adding the first perturbation to form a negative example is straightforward. But why feeding H^{-l_t} into the decoder and then adding another small perturbation make it a positive example?

Question C: I find it quite surprising that BT (back-translation) is not helping in most cases for both high-resource and low-resource languages. Do you have any intuition or observation for this?

Question D: I find one ablation experiment a bit hard to interpret: in Table 4, the 2nd ablation of #1 that uses AE_fwd only. One side note: the name, backward (or forward) autoencoder, is a bit confusing because they are not real autoencoders. For example, for your backward autoencoder, its input is X_{l_t}, and its output is X_{l_t}. For a real "autoencoder", it needs to reconstruct its input. So putting your backward and forward autoencoder together forms one real "autoencoder". Your backward autoencoder is the "encoder" and your forward autoencoder is the "decoder". From my understanding, if you train the model with AE_fwd loss only, it will soon learn to copy input (target sentence). Hence, the AE_bkd is critical for learning meaningful hidden presentation Z_{bkd}. But in Table 4, using AE_fwd only surprisingly works quite well.

Question E: In lines 882 to 884, it says 128 sentences are randomly sampled to compute I1 and I2 scores. I wonder why only 128 examples are used and how different the results will be if you sample two different sets of 128 examples.

Question F: In Figure 5, Example 1 shows copy problem. My understanding of "copying" is copying words from the source sentence. However, none of the highlighted red errors are copied English tokens from the source. Do you mean other "copying" problems?

**Reasons To Accept:**

1. Data imbalance and representation degeneration are two important problems to address for multilingual translation.

2. Experiments are thoroughly done in this paper and demonstrate the effectiveness of Bi-ACL framework.

**Reasons To Reject:**

1. The method section (Section 3) is not easy to follow. In particular, I have a hard time understanding how contrastive learning is performed and why the positive (or negative) example is considered positive (or negative). The notations are confusing. Please refer to my questions for more details.


**Reproducibility:**

3: Could reproduce the results with some difficulty. The settings of parameters are underspecified or subjectively determined; the training/evaluation data are not widely available.

**Reviewer Confidence:**

3: Pretty sure, but there's a chance I missed something. Although I have a good feel for this area in general, I did not carefully check the paper's details, e.g., the math, experimental design, or novelty.

---

> ### Author Rebuttal · Authors · 2023-08-28
>
> We thank the reviewer for their insightful feedback.
>
> Q1: Questions in Eq. 6?
>
> A1: We apologize for any confusion caused by our description of the formula. In the camera-ready version, we will provide a more detailed explanation to avoid any misunderstanding. We have also noticed some typos, such as H^{+l_t} in main text should be changed to H^{+l_s} as shown in Eq. 6. Since we are translating from the target language to the source language, our positive example is the source language. Additionally, in the last formula of Eq. 6, it should be ‘Decoder’ instead of ‘Encoder’. We will also map the formula to Figure 2 to provide a better understanding of our method.
>
> Q2: Questions on the positive example construction?
>
> A2: Due to space constraints, we have provided more details on the construction of positive and negative examples in Appendix A. The traditional contrastive learning method [Pan et al., 2021] uses ground-truth examples as positive examples. However, in our scenario, the ground-truth examples are noisy due to the absence of a parallel corpus as a supervised signal. To minimize the impact of this noise on our model, we have added perturbations on the decoder side to ensure that the constructed positive examples are closely aligned in the representation space.
>
> Q3: Explain the reasons why BT does not improve the performance both in high-resource and low-resource languages?
>
> A3: In low-resource languages, the performance of the language pair and its reverse direction in the original m2m_100 model is extremely poor, nearly zero. As a result, using back-translation (BT) leads to inferior performance compared to m2m_100. For high-resource languages, the language pairs already perform well in the original m2m_100 model, making it difficult to demonstrate that incorporating additional pseudo-parallel data will yield better results than not using the pseudo-corpus at all. Another potential issue is that the large amount of monolingual data we use, and the substantial amount of pseudo-parallel data we obtain from BT, may interfere with the pre-trained model. This is consistent with the findings of Liao et al., 2021 [Back-translation for Large-Scale Multilingual Machine Translation].
>
> Q4: Explain Table 4 ?
>
> A4: Yes, combining the backward and forward processes results in a traditional autoencoder. We have split it into two directions to provide a better understanding of our method. However, the backward part is also a complete encoder-decoder process, as detailed in Eq. 2 and Eq. 4, as well as in Figure 2. Therefore, using only AE_bkd will also work.
>
> Q5: Why only 128 examples are used in computing the I_{1} and I_{2} score?
>
> A5: We have experimented with larger datasets and obtained similar results. We chose to use 128 samples to expedite the calculation of these two values, as they are performed on the CPU. In the camera-ready version, we will update the appendix with results from different sample sizes.
>
> Q6: Copying problem in Figure 5?
>
> A6: We apologize for any confusion and thank you for bringing this to our attention. Yes, we were referring to the repetition problem, as described by Fu et al., 2021 [A Theoretical Analysis of the Repetition Problem in Text Generation], not the copy problem. We will make the necessary corrections in the camera-ready version.

---

### Official Review · Reviewer_LrBM · 2023-08-04

**Soundness:** 4

**Excitement:**

4: Strong: This paper deepens the understanding of some phenomenon or lowers the barriers to an existing research direction.

**Paper Topic And Main Contributions:**

This paper addresses the issue of translating into low resources language for multilingual machine translation which are data imbalance and representation degeneration problems using Bi-Acl (Bidirectional autoencoder and contrastive learning) framework that can leverage monolingual corpus and dictionary. BiACL works by first creating online pseudocorpus using constrained decoding with dictionary as helper. This corpus is then used for training the forward and backward autoencoder (src->trg, and trg->src). Contrastive loss is added on top of the autoencoder for the model to distingusih between positive and negative examples.



**Questions For The Authors:**

- Q1: I don't understand how the contrastive learning works in your case. Given batches of pseduoparallel pairs, did you calculate both losses at the same time (contrastive and autoencoding)? I need a bit more clarity here.
- Q2: In the introduction the (real) low resource dictionary seems available. This might be very useful given dictionary construction often requires parallel text that is hard to find. Did you do any experiment using real world dictionary?
- Q3: I want to understand the effectiveness of the method when we have more data. In a real world scenario, you often combat the low resource language just by collecting more data. How much data do we need so that standard cross entropy training is better than your proposed method?
- Q4: Why we need to use \lambda and (1-\lambda) for Equation 10. Do these losses behave like probability? Why can't we use two hyperparameters for them?

**Reasons To Accept:**

- Although BLEU score is low, but this is a very low resource language, where baseline model BLEU is close to 0. Experiments showed respectable gains across low, medium, and high resource language. This showed robustness of the method.
- Works on tail very low resource language is not common. The paper raises the bar of these languages.

**Reasons To Reject:**

- The intuition behind contrastive learning is hard to follow. The paper jumps into explaining the technical details of the method before explaining the intution behind it.

**Reproducibility:**

3: Could reproduce the results with some difficulty. The settings of parameters are underspecified or subjectively determined; the training/evaluation data are not widely available.

**Reviewer Confidence:**

3: Pretty sure, but there's a chance I missed something. Although I have a good feel for this area in general, I did not carefully check the paper's details, e.g., the math, experimental design, or novelty.

**Typos Grammar Style And Presentation Improvements:**

The notations are hard to follow, notably indices are too small to read and quite cluttered. These are my suggestion:
- l_{s} can be changed to 'f' (french or source language, this is common convention)
- l_{t} can be changed to 'e' (english or target language, this is common convention)
- fwd, bkd can be changed to use arrowhead.
- tt and ss can be changed to T and S or |F| and |E|
- Using + and - for marking positive and negative examples are quite confusing because those can be interprated as unary operator for exponent.
- Equation 7 and 9 can use the exp() function, even better we can use log exp(...) - log sum exp(...) because it is fraction inside logarithm for clarity.
- Figure 2 needs more captions and explanation. It's not obvious for the readers to understand what's going on there.

---

> ### Author Rebuttal · Authors · 2023-08-28
>
> We thank the reviewer for their thoughtful comments and suggestions.
>
> Q1: How does contrastive learning work?
>
> A1: Yes, the four losses presented in this paper are calculated simultaneously in each batch.
>
> Q2: Did you do any experiment using a real world dictionary?
>
> A2: Yes, we ran experiments using a real world dictionary. While it is possible to extract bilingual dictionaries from most language pairs in Panlex, the dictionaries are often of low quality or contain fewer than 100 pairs. For this reason, we choose to use higher quality Wiktionary dictionaries and obtain bilingual dictionaries between non-English pairs by using English as a pivot language. For the experiments using a real world dictionary (e.g., Panlex), you can refer to Table 8 in Appendix.
>
> Q3: How much data do we need so that standard cross entropy training is better than your proposed method?
>
> A3: We appreciate your suggestion and thank you for bringing this to our attention. While we haven't run the experiment you propose for this paper, we will make sure to explore it in future work.
>
> Q4: Why can't we use two hyperparameters for them in Eq. 10?
>
> A4: Yes, the weights in our model can be interpreted as probabilities. We are aware of related work that dynamically adjusts these weights and plan to explore the application of dynamic weights to our work in the future to see if it leads to further improvements.

---

### Official Review · Reviewer_G7Tr · 2023-08-05

**Soundness:** 3

**Excitement:**

3: Ambivalent: It has merits (e.g., it reports state-of-the-art results, the idea is nice), but there are key weaknesses (e.g., it describes incremental work), and it can significantly benefit from another round of revision. However, I won't object to accepting it if my co-reviewers champion it.

**Paper Topic And Main Contributions:**

The paper proposes a new approach called  Bi-ACL to fix two problems in multilingual NMT models, including data imbalance and representation degeneration.

Main contributions:
- a novel approach that uses only target-side monolingual data and a bilingual dictionary to improve MNMT translation quality;
- designed bidirectional autoencoder and bidirectional contrastive learning, to address the data imbalance and representation degeneration problems;
-  the proposed approach has zero-shot domain transfer and language transfer capabilities;

**Questions For The Authors:**

- Could the authors add statistical significance tests to the BLEU tables? Because the BLEU scores in Table 1 are very low. It is not very clear how much the BLEU improvements imply.

**Reasons To Accept:**

- the proposed approach is novel;
- the paper is well written;

**Reasons To Reject:**

- the BLEU scores reported in Table 1 are very low, so it may not be very clear how much the BLEU improvements really imply;

**Reproducibility:**

3: Could reproduce the results with some difficulty. The settings of parameters are underspecified or subjectively determined; the training/evaluation data are not widely available.

**Reviewer Confidence:**

3: Pretty sure, but there's a chance I missed something. Although I have a good feel for this area in general, I did not carefully check the paper's details, e.g., the math, experimental design, or novelty.

**Typos Grammar Style And Presentation Improvements:**

- Line 112: "MNT";

---

> ### Author Rebuttal · Authors · 2023-08-28
>
> We thank the reviewer for helpful comments.
>
> Q1: add statistical significance tests to the BLEU tables?
>
> A1: Thank you for your valuable feedback. We agree that it is important to add statistical significance tests to our experiments. We used the sacrebleu tool for significance testing and the results show that the difference between the baseline systems is not large, as the BLEU score between them is almost 0 and has strong randomness. However, the difference between our method and the baseline system is large, indicating that our improvement is meaningful despite the low BLEU score. Additionally, our case study (Figure 5 in the appendix) found that the baseline system exhibits a serious repetition phenomenon, where the generated translation repeats previously generated words. Our method successfully addresses this problem, further confirming that our approach is quite different from the baseline method. Although we are unable to add experimental results during the rebuttal stage, we will ensure that this part of the significance test will be included in the camera-ready version to address any concerns.

---

### Meta-Review · Area_Chair_8BSU · 2023-09-19

**Recommendation:** 3

**Metareview:**

The paper introduces a novel framework named Bi-ACL, aimed at addressing two key issues in multilingual Neural Machine Translation (MNMT), specifically data imbalance and representation degeneration.

Pros:
- Combine several popular (and easy to implement) techniques into a novel framework for enhancing MNMT. These improvements demonstrate consistency across various data conditions.
- Utilizes readily available resources that are more accessible than parallel data, namely target-side monolingual corpora and bilingual dictionaries.
- Addresses the challenge of extremely low-resource translation directions (alongside with mid- and well-resource).
- The paper is presented in a clear and comprehensible manner.

Cons:
- The rationale behind the combination of these techniques is not sufficiently elucidated. The individual techniques themselves are not novel.
- Some minor ambiguities/errors in the paper require clarification/correction, which was addressed during the rebuttal phase thanks to the reviewers' feedbacks.

---

### Decision · Program_Chairs · 2023-10-07

**Decision:**

Accept-Findings

**Comment:**

The paper introduces a novel framework named Bi-ACL, aimed at addressing two key issues in multilingual Neural Machine Translation (MNMT), specifically data imbalance and representation degeneration.

Pros:
- Combine several popular (and easy to implement) techniques into a novel framework for enhancing MNMT. These improvements demonstrate consistency across various data conditions.
- Utilizes readily available resources that are more accessible than parallel data, namely target-side monolingual corpora and bilingual dictionaries.
- Addresses the challenge of extremely low-resource translation directions (alongside with mid- and well-resource).
- The paper is presented in a clear and comprehensible manner.

Cons:
- The rationale behind the combination of these techniques is not sufficiently elucidated. The individual techniques themselves are not novel.
- Some minor ambiguities/errors in the paper require clarification/correction, which was addressed during the rebuttal phase thanks to the reviewers' feedbacks.